# Effects of Pea (*Pisum sativum* L.) Cultivars for Mixed Cropping with Oats (*Avena sativa* L.) on Yield and Competition Indices in an Organic Production System

**DOI:** 10.3390/plants11212936

**Published:** 2022-10-31

**Authors:** Lina Šarūnaitė, Monika Toleikienė, Aušra Arlauskienė, Kristyna Razbadauskienė, Irena Deveikytė, Skaidrė Supronienė, Roma Semaškienė, Žydrė Kadžiulienė

**Affiliations:** Lithuanian Research Centre for Agriculture and Forestry, Akademija, LT-58344 Kėdainiai, Lithuania

**Keywords:** intercropping, field pea, organic agriculture, sowing rate

## Abstract

The benefits of cereal-legume mixed cropping is a sustainable agricultural practice. However, knowledge of the genotypic differences of semi-leafless pea varieties is not enough to help them compete with cereals. In this study, the effects of Lithuania’s newest Pisum sativum cultivars (‘Egle DS’ and ‘Lina DS’) and, for comparison, a control cultivar (‘Jūra DS’) established with *Avena sativa* in mixed cropping system were investigated. Three years of field trials (2018, 2019 and 2020) with four experiments involved three different mixtures of each field pea cultivar with oat. The aboveground biomass of mixed cropped new field pea cultivars was found to be significantly higher: biomass of cultivars ‘Egle DS’ increased by 17.0% and ‘Lina DS’ by 7.2% on average compared with the control cultivar ‘Jūra DS’. For the mixed cropping system, statistically greater total aboveground biomass was observed with plant ratios of 50% pea + 50% oat and 60% pea + 40% oat compared to peas monocultures. Mixed cropped oat was the dominant species in all tested mixture compositions; however, the highest total grain yield of mixed crops was obtained when new pea ‘Lina DS’ and ‘Egle DS’ cultivars were included in the mixtures compared with the control cultivar. The new pea cultivar ‘Egle DS’ had a greater effect on protein content compared to other tested pea cultivars. In the new pea cultivars ‘Lina DS’ and ‘Egle DS’, the higher photosynthetic capacity and aboveground biomass of mixed cropped pea with oat showed mixture effects in the mixed cropped system and could increase total yield compared with pea monoculture. Generally, the new pea cultivars displayed a greater Land Equivalent Ratio (LER) value, resulting in the greatest yield among the mixtures on average for all three years and all four experiments. Future research could optimize the effects of pea cultivar mixtures with cereals to further improve the yield of organic mixed cropping systems.

## 1. Introduction

Mixed cropping is a sustainable agricultural practice used all over the world to make a very efficient use of resources [1]. Most of the studied is legumes and cereals mixed cultivation systems [2,3,4]. This leads to improved utilisation of environmental resources such as light, water and nutrients [5,6,7,8]. The ability of legume crops to complement the N–soil system is a clear benefit that depends on crop systems to maintain the soil nitrogen supply at sustainable productivity levels [9,10,11]. Grain legumes cultivated in mixed cropping systems with cereal include pea, vetch, soya, lupin and bean [12,13,14,15]. The cultivation of nitrogen-fixing legumes in mixed cropping systems can also improve the content of organic carbon in the soil and the availability of phosphorus, which are the main factors for soil fertility [16]. Cereal -legumes intercrops improve soil conservation [17,18] and yield stability [19,20] are favourable for controlling weeds, pests and diseases [21,22,23,24]; and improve the quality of cereal grains [25]. Thus, this intercropping meets not only the need to reduce chemical inputs (fertilizers and phytosanitary products) and their associated production costs but also protects the environment [4]. Many intercropping studies focus on variables such as the relative plant population of the constituent crops, planting dates, treatment with fertilizers and the spatial orientation of the crop, etc. [1,26,27].

Mixed cropping systems are focused on the ecological principles of competition, complementarity, and facilitation [28]. The components of the intercropping crop are closely related and interact with each other [1,29]. If interspecific competition for crop yield factors is lower than intraspecific competition, these crop species can share just a part of the same land area and the reduced competition principle is in action [30]. An important aspect of the intercropping system is the scale of competition between crops. In the face of greater competition, the interaction between the genotype and agronomic practices is very important [29]. In cereal and semi-leafless pea mixed cropping systems, cereals are more competitive than peas, therefore the yield most often are decreases in mixed cropping system at the expense of pea yields [31]. Similar results are given by other researchers [26,32]. In mixed cropping system less, competitive semi-leafless peas grow in the shade of tall cereals. It is claimed that the shadow adversely affects photosynthesis, getting into the roots of assimilates, fixation of N and yield [32,33]. Such varietal indicators as leaf area, content and quality of chlorophyll, photosynthetic capacity can increase shade tolerance [33].

To reduce the negative competition between crop species, it is necessary to explore the cultivation of the strongest new varieties of legume crops that have simpler requirements for growth conditions [34], stronger competencies for nutrients and higher, more stable grain yields [35,36]. Studies have shown that different pea varieties, harvested from two subsequent years at various locations, produced grain with a different general chemical composition [37]. Moreover, field pea varieties can respond differently to high growth temperature, revealing their diverse potential to resist heat stress [38]. Adapted field pea breeding lines have been developed that combine the ability to grow under sustainable conditions with other desirable agronomic traits with maintaining adequate productivity [39]. An investigation of pea varieties in a mixed cropping system showed that variety diversification may increase yield and promote microbial interactions by affecting the soil, plant health and broader ecosystem functions [40]. However, selecting individual plant varieties and different species for mixed cropping at different proportions might influence and reduce intraspecific competition. The selection of field pea genotypes with comparable phenology but contrasting stature and growth when intercropped with cereals in breeding programmes helped to determine pea proportion, height, and yield in dual crops, indicating the strict association of pea stature with yield and competitive ability [6,41].

The morphological and physiological features of new breeds in relation to the tolerance and avoidance of the shadow, such as the height, density is important [33]. However, not enough is known about the strategy that peas can use when competing with cereals.

Mixed cropping systems most often grown on low-yielding or low-cost farms. But despite the potential benefits due to the additional workforce and more complex management, mixed cropping is not widely used in modern, mechanized grain cultivation systems [42]. The implementation of the Green Deal program, reducing the use of fertilizers, pesticides, energy costs will stimulate (in part) the cultivation of mixed crops. Therefore, improving the practice of growing mixed crops will make these crops more attractive and at the same time gain greater confidence from growers. The present investigation was carried out with the objectives to quantify genotypic differences between crop species and their most suitable combination for mixed cropping in an organic farming system.

## 2. Materials and Methods

### 2.1. Experimental Sites

The study was carried out at the Lithuanian Research Centre for Agriculture and Forestry, situated in Central Lithuania’s lowland region in three subsequent years (2018–2020) via four experiments: 2018, 2019 and 2020 at the Akademija site (55°24′ N, 23°51′ E) and 2020 at the Tiskūnai site (55°348′ N, 24°15′ E). The soil of the Akademija experimental site is an *Endocalcari–Endohypogleyic Cambisol* (CMg-n-w-can) with a loam texture. The topsoil (0–25 cm) had a pH of 7.5, 74–79 mg·kg^−1^ available P_2_O_5_, 2.3% humus content and 135–140 mg·kg^−1^ potassium. At the experimental site in Tiskūnai, the topsoil had a pH of 7.9, 81–94 mg·kg^−1^ available P_2_O_5_, 1.5% humus contents and 420 mg·kg^−1^ potassium. In the region of the experiments, the total annual rainfall and average temperature are 570.1 mm and 6.5 °C, respectively.

Weather data were collected at a stationary meteorological station located in Akademija using temperature and rainfall sensors (Figure 1). In 2018, the temperature of the growing season was higher than the average. In 2019, the temperature was higher than the perennial temperature during the vegetative phase and harvesting, but the precipitation was very uneven. In 2020, the temperature and precipitation decreased during the sowing period but increased during the vegetative phase.

### 2.2. Experimental Design and Plant Material

In this study, two species (field pea and oat) were planted as monocrops and mixtures. For the experiment, two new cultivars of semi-leafless field pea Egle DS and Lina DS, and cultivar one old cultivar Jūra DS, one oat Viva DS were used. The experimental plots were laid out in a complete one-factor randomized block design with four replications. The individual plots had an area of 15 m^2^ (10 m × 1.5 m) with 12.5 cm spacing between rows. Three different mixtures of each field pea (*Pisum sativum* L.) cultivar and oat (*Avena sativa* L.) (50% pea + 50% oat (1:1), 60% pea + 40% oat (3:2), 70% pea + 30% oat (7:3)) and monocultures of the three field pea cultivars (100% pea (1:0)) and pure oat (100% oat (1:0)) were used as the experimental treatments. The intercrop design was based on the proportional replacement principle, with mixed pea seeds and oat seeds planted at the same depth in the same rows at the appropriate proportion of pea and oat. The oat seed rate was 6.0 mln. seeds ha^−1^ and that of pea was 1.0 mln. seeds ha^−1^ for the monocrop treatments. The following crop cultivars were used: standard field pea (cv. Jūra DS) with a medium vegetation period (91 days) and a plant height of 90.4 cm; the new field pea cultivar cv. Egle DS with a longer duration of vegetation (94 days) and a plant height of 98.9 cm; the new field pea cultivar cv. Lina DS with a medium vegetation period (91 days) and a plant height of 85.6 cm; and oat (cv. Viva DS) with a short vegetation period (85 days) and a plant height of 106 cm. During the experimentation, chlorophyll content was measured in leaf tissues at different vegetative stages, by non-destructive SPAD method, using SPAD 502 Plus Chlorophyll Meter [43].

Plants were harvested by a harvesting machine in all experimental plots. Before harvesting, plants were manually selected by cutting at the soil surface at full ripeness in 1.0 m^2^ sampling plots (0.25 m × 0.25 m) in 4 places for productivity analysis. Plant samples were divided into the component species, which were then used to determine yield components including grain and straw yield and 1000-kernel weight.

### 2.3. Expected Grain Yields vs. Actual

The combined total grain yield was calculated for all the mixture, and yield was calculated separately for pea and oat in the mixtures. The value was defined by the difference between actual grain yield and expected yield. For this, expected yield was calculated from the monocrop multiplied by the plant ratio used in the mixtures.

The *Land Equivalent Ratio* (LER) was calculated using the equation proposed by Wiley and Rao (1980) [44]. It denotes the relative land area under a monocrop required to give the same yield as that obtained under a mixed or an intercropping system at the same level of management [45]. The equation includes the pure yield of the main crop (Y_aa_), the yield of the intercrop (Y_bb_), and the proportion of the main crop (Y_ab_) and the intercrop (Y_ba_) (1).
(1)$$\mathrm{LER}=\frac{Y_{\mathrm{ab}}}{Y_{\mathrm{aa}}}+\frac{Y_{\mathrm{ba}}}{Y_{\mathrm{bb}}}$$

### 2.4. The Relative (k) Crowding Coefficient

The coefficient (k) gives the yield per unit of area for calculating the ‘expected’ yields on the basis of how much of the area is initially allocated to each crop [44]. In this equation, k_ab_ is the relative crowding coefficient of Crop a intercropped with Crop b, Y_ab_ is the yield per unit area of Crop a intercropped with Crop b, Y_aa_ is the yield per unit of area of pure Crop a, Z_ab_ is the proportion of the intercropped area initially allocated to Crop a and Z_ba_ is the proportion of intercropped area initially allocated to Crop b.
(2)$$k_{\mathrm{ab}}=\frac{Y_{\mathrm{ab}}}{Y_{\mathrm{aa}}-Y_{\mathrm{ab}}}\times \frac{Z_{\mathrm{ba}}}{Z_{\mathrm{ab}}}\;$$

### 2.5. (A) Aggressivity Value

Aggressivity(A) was calculated using the equation proposed by McGilchrist (1965) [46] (3). It measures the relative yield increase or decrease in Component a growing together with Component b. The aggressivity could be calculated for both crops in the mixture.
(3)$$A=\frac{\mathrm{Actual}\;\mathrm{yield}\;\mathrm{of}\;a\;\mathrm{when}\;\mathrm{intercropped}}{‘\mathrm{Expected}’\;\mathrm{yield}\;\mathrm{of}\;a\;\mathrm{when}\;\mathrm{intercropped}\;}-\frac{\mathrm{Actual}\;\mathrm{yield}\;\mathrm{of}\;b\;\mathrm{when}\;\mathrm{intercropped}}{‘\mathrm{Expected}’\;\mathrm{yield}\;\mathrm{of}\;b\;\mathrm{when}\;\mathrm{intercropped}\;}$$

### 2.6. Competitive Ratio

The competitive ratio (CR) gives the yield per unit of area, calculating the ‘expected’ yields on the basis of how much of the area is initially allocated to each crop [44] (4).
(4)$${\mathrm{CR}}_{a}=\frac{Y_{\mathrm{ab}}}{Y_{\mathrm{aa}}\times Z_{\mathrm{ab}}}÷\frac{Y_{\mathrm{ba}}}{Y_{\mathrm{bb}}\times Z_{\mathrm{ba}}}$$

### 2.7. Statistical Analysis

Statistical analysis was performed using SAS software version 9.4 (SAS Institute Inc., Cary, NC, USA, 2002–2010). Homogeneity and normality were verified using Bartlett’s test. Interactions between years were tested and no significant interactions were found. Experimental data were analysed by one-way analysis of variance (ANOVA), and mean comparisons between treatments were performed using Duncan’s means separation test. The least significant difference was calculated using a probability level of *p* < 0.05.

## 3. Results

Crop quality and productivity had no significant effect in the three cropping seasons, but the effect of the mixing crop system × field pea cultivar interaction on 1000-kernel weight was found to be significant (*p* < 0.001). On average across the three-year experimental period, the oat mixed with field pea had a significantly higher protein concentration in grain compared with oat grown as a monocrop (Table 1). In the mixtures, we found that the tested field pea cultivars improved different quality parameters of oat compared with that of the monoculture. The new pea cultivar Egle DS had a greater effect on protein content, ranging from 13.1% up to 13.4% in mixtures with sowing ratios of 3:2 and 7:3, respectively.

On average, the protein content in the grains of pea cultivars grown as a monocrop was significantly higher (<0.001) for the new cultivar Egle DS compared with the control cultivar Jūra DS and second new cultivar Lina DS. The protein concentration in oat grain was significantly higher under mixed cropping with pea than in the oat monoculture crop. The maximum protein concentration in oat was found for the Viva DS and Egle DS mixed crop at a ratio of 60% pea + 40% oat (3:2). Other mixed cropping treatments showed marginally lower crude protein concentration in oat grains.

On average across the experimental period, the pea 1000-kernel weight was significantly lower for the new cultivars compared with the control cultivar Jūra DS grown as a monocrop.

However, the new cultivar Egle DS showed essentially higher productivity in terms of 1000-kernel weight in mixtures with oat as well as protein concentration in the grains. There was a significant difference in plant height among the different mixed treatments on average during the three growing seasons (Table 2).

The height of field pea mixed with oat was significantly greater for the new cultivar Egle DS compared with the control cultivar Jūra DS on average (Table 2). In general, the plant mixture ratio had a significant effect (*p* < 0.001) on field pea height regardless of what cultivar was incorporated in the mixture. All cultivars had a lower height in mixtures compared with field pea sown as a monoculture. For oat, the mixture ratio configurations had no significant effect on plant height compared with oat grown as a monoculture. The mixture ratio significantly influenced the oat grain number per ear, and there was significant difference between the mixtures with pea and oat grown as a monoculture. The new field pea cultivars Egle DS and Lina DS showed significantly greater effects on oat productivity and quality parameters compared with mixtures with the control pea cultivar Jūra DS or oat grown as a monoculture.

Th aboveground total biomass of field pea and oat was significantly influenced by the mixture ratio treatment with the control cultivar Jūra DS on average during three growing seasons (Table 3). The biomass of pea in mixtures was significantly increased (*p* < 0.001) in the two new cultivars Egle DS and Lina DS. The new field pea cultivar Lina DS was observed to have lower aboveground biomass, which varied from 128 to 181 g m^−2^ but showed no significant difference from mixtures with the cultivar Egle DS, where pea biomass varied from 168 to 250 g m^−2^ depending on the pea ratio in the mixtures with oat. The aboveground biomass of new field pea cultivars in mixed cropping was found to be significantly higher for Egle DS by 17.0% on average and for Lina DS by 7.2% on average compared with the control cultivar Jūra DS. For the mixed cropping system, greater total aboveground biomass was observed with plant ratios of 50% pea + 50% oat (1:1) and 60% pea + 40% oat (3:2).

Field pea grown as monoculture showed higher SPAD values compared with the mixed cropping treatments, and a significant difference was found between the new pea cultivars Egle DS and Lina DS, and the control cultivar Jūra DS (Figure 2). The SPAD value of the field pea monoculture with Egle DS was 12.3% lower and that with Lina DS was 5.8% lower than that with the control cultivar Jūra DS on average at the beginning of the vegetation period (June 2). For oat, all mixed cropping treatments had no significant difference in the SPAD value compared with oat grown as monoculture at the same growing stage (Figure 3). In the later vegetation period, the SPAD value of oat in the mixed cropping system was found to be significantly different from oat grown as monoculture. On 8 July, the SPAD values were found to be the highest in oat mixed with all tested pea cultivars at a ratio of 7:3; however, the new cultivar Lina DS significantly influenced the SPAD value at all ratios, but the greatest effect was found for the ratio 50% pea + 50% oat (1:1).

The grain yield did not differ significantly among the treatments (*p* = 0.068) in 2018 (Table 4). The lowest yields were found in the mixtures with the new pea cultivar Jūra DS, the highest yield was found for the new cultivar Egle DS.

Lina DS was the best pea grown as a as monoculture. In 2019, the treatments in which pea was grown as a monoculture had significantly lower grain yields (*p* <0.001) and the monoculture of oat had the greatest. If we compare the mixed cropping systems, the treatments with the pea variety Egle DS had the highest yields. Oat and pea intercrops proved more productive than monocultures of pea. The differences in production, however, between the mixed cropping and pea monoculture proved to be smaller than expected. In 2020, the monocrop of field pea varied between the different sites, but the trends were determined to be similar. Jūra DS and Lina DS produced near two times lower grain yields at Tiskūnai compared with those at Akademija. Generally, the highest grain yield was obtained when the new pea cultivar Egle DS was grown together with oat, where the higher yield of the mixed cropping treatment was due to the oat component. The difference between actual total mixed cropping grain yield and the yield expected showed that the pea cultivar Egle DS increased the total mixed grain yield compared with the other pea cultivars (Figure 4), especially when the ratio in the mixture of Egle DS was 60% and that of oat was 40% (3:2).

However, the grain yield of pea in the mixed cropping systems showed was higher with the new cultivar Lina DS (Figure 5). However, oat grain yield decreased when field pea yield was higher in the mixed cropping systems (Figure 6).

The highest oat grain yield in the mixed cropping treatments was obtained when the ratio of oat was 50% in the mixture with pea Egle DS (Table 5). The LER of grain yields was generally below unity except, mixtures with control cultivar Jūra DS and new cultivar Egle DS at 7:3 rate, indicating the lower grain productivity of mixed cropping systems with the control pea cultivar Jūra DS (Table 5). Relatively, the crop of mixed intercrops resulted in in LER values > 1 for grain yields. Mixed intercrops with a higher share of oat in the mixtures with Jūra DS and Egle DS showed a trend towards lower LER values for grain yields. Only the three-year mean of the control cultivar Jūra DS was less than one. The component crops did not exhibit equal competitive intensity based on aggressivity. In the three sowing years, the aggressivity index (A) of oat relative to field pea’s A value was positive in all the mixed cropping treatments. Furthermore, the average A values of the three years were significantly greater than zero (*p* = 0.0646), indicating that oat was the dominant species and had much greater competitiveness in the mixed cropping system of oat with field pea, especially when oat was mixed with the pea cultivar Egle DS. The interspecific competitive abilities were determined by the relative crowding coefficient (k). Regarding the k values of all mixed cropping systems, on average across the three years, the k value of oat was always greater than the k of pea (Table 5). However, at the 1:1 sowing ratio with the new pea cultivar Lina DS, the k value was greater than that of the other tested field pea cultivars at the same sowing ratio. The new pea cultivar Lina DS was more competitive than Jūra DS and Egle DS. Similar results were observed for the competitive ratio (CR). The CR value in different oat–pea mixed cropping systems always exceeded 1.0 on average for all three years and all four experiments, and thus were higher than the competitive ratios of all field pea cultivars (Table 5). The average CR value of oat over the study period was also higher than 1.0 for each mixed cropping configuration. In contrast, the average field pea CR values were less than 1, suggesting that oat had greater competitive intensity relative to field pea in all tested combinations. However, the new pea cultivar Lina DS mixed with oat utilized the resources more aggressively than the other tested pea cultivars and its CR values were significantly higher compared with those of the cultivars Jūra DS and Egle DS (*p* = 0.0813).

## 4. Discussion

Determining the crop yield indicators involved in growing mixes is the most important component in developing at mixed cropping system. The results reported by Ghaley et al. (2005) [14] showed that fertilization with nitrogen improved the competitive ability of mixed crop due to increases in aboveground dry matter, grain and straw. In eastern Austria, growing oat–pea mixed crops did not achieve a greater grain yield than monocultures of these crops on fertile soil. Oat was the dominant crop in the mixtures and strongly overcame pea, especially with additional N fertilization [47]. Lauk and Lauk (2008) [48] found that when growing pea–oat in mixes, the higher yield of the mixed cropping system was more productive than that of monoculture crops. However, it has been established that mixed cropping reduces the yield of the separate components compared with the yields of the individual crops as monocultures [21]. Numerous researchers have presented theories and models that confirm the total yield stability in mixed cropping systems. Mead and Willey (1980) [49] analysed mixed cropping systems in detail and found that such systems’ crop yields are more stable. Research into modern agricultural practices tend to involve more crop cultivars in mixed cropping due to the plants’ root microbial community composition in comparison with cropping with a single cultivar [40]. Our results suggest that mixed cropping of oat with different field pea cultivars produced varying grain yields. Regardless of the crop ratio in mixtures and the field pea cultivar, the competitive ability was different and the new pea cultivars may be more suitable to be grown in different systems: Egle DS was suitable for monocropping, but Lina DS was superior in an organic mixed cropping system. New field pea cultivars mixed with oat achieved the greatest mixed cropping advantage, which was mainly influenced by the effects of biological nitrogen fixed by pea on the higher concentration of crude protein in oat grain compared with the oat monocrop. The higher CR values of the pea cultivar Lina DS in our study indicated that on average, this cultivar was more competitive than the other tested pea cultivars at different sowing ratios in the mixtures. Oat production was the major factor that determined the overall yield compared with the yield of the pea monoculture. On average across the three experimental years at two sites, the highest crop grain yield was found for the oat monoculture. However, the concentration of crude protein in the oat grain was higher in oat grown in a mixture with field pea. The results of a meta-analysis suggested that mixed cropping consistently stimulates complementary nitrogen use between legumes and cereals [50]. Other researchers have also indicated that mixed cropping improves the stability of crop production and provides greater crop security and quality [51,52]. Furthermore, mixed cropping systems are a preferred land-use system to compensate for the disadvantages of arable land [53].

It has been reported that short strong-strawed pea cultivars appear to be unsuitable for mixtures with oat. Mixtures with long-strawed pea cultivars were more successful and increased the competitive ability of pea in the mixed crop system [54]. As pea plants grown in a mixed crop system are shaded by oat, this may have contributed to the reduced TKW [47]. Typically, cereals are established as the dominant crop in a cereal–legume mixed cropping system [55,56,57]. It has also been found that oats have outstanding tillering ability and peculiarities of nutrition: they grow tall and therefore their competitive characteristics are strong [31,58]. A significant change in productivity assumes that mixed crops are possible when the morphological characteristics of the two mixed crops are different [59]. Previous studies have also reported that this complementarity exists due to the different crop species such as *Panicum miliaceum* L. and *Vigna radiata* L. with morphological differences in the root systems and crop heights [60]. The new pea cultivars Egle DS and Lina DS investigated here are long-strawed semi-leafless varieties and, unlike the control pea cultivar Jūra DS, increased the competition with oat in a mixture, and the 1000-kernel weight of pea was found to be similar to that of the new pea cultivars grown as a monoculture.

Agricultural practices such as cultivation modify the soil’s physical and chemical properties and, consequently, change the soil nutrient conditions and enzyme activities, including soil nitrogen [50]. Therefore, meteorological factors such as precipitation during the growing season are very important. Higher precipitation in the growing season is more favourable to the growth of microorganisms and produces more nutrients [61]. Our results revealed an increase in crop productivity in the mixed cropping systems at both experimental sites in 2020, when the amount of precipitation was highest during the vegetative growth stage compared with previous years. Increased competitive properties of field pea with oat were observed in mixtures of new field pea cultivars and oat. Our results support previous researchers’ results showing that various factors such as crop density, design and mixed crop composition should be considered to regulate the interaction between diverse crop species and maximize crop growth in mixed cropping systems [62]. Rodriguez et al. (2020) [28] reported the results of a meta-analysis that showed a great opportunity to improve the efficiency of soil nitrogen usage in global crop growing systems by mixing cereals and grain legumes, due to the sharing of the mineral nitrogen because of the competitive ability of the cereal for mineral nitrogen while also stimulating symbiotic nitrogen fixation in the legume crops.

## 5. Conclusions

Mixed cropping with field pea and oat can lead to grain yield advantages when the proportions and crop densities are at the optimal ratio (3:2). The proportion of pea can be high, depending on the field pea cultivar’s architecture. Therefore, new long-strawed pea cultivars mixed with oat could be an efficient approach for organically grown grain production in mixed cropping systems. In the new pea cultivars Lina DS and Egle DS, the higher photosynthetic capacity and aboveground biomass of a mixed crop of pea and oat showed that mixture effects in a mixed cropped system can increase the total yield compared with a pea monoculture. Generally, the new pea cultivars Lina DS and Egle DS displayed a greater LER value, resulting in the greatest yield of the mixtures on average for all three years and all four experiments. Future research may optimize the effects of pea cultivar mixtures with cereals to further improve the yield of organic mixed cropping systems.

## Figures and Tables

**Figure 1 plants-11-02936-f001:**
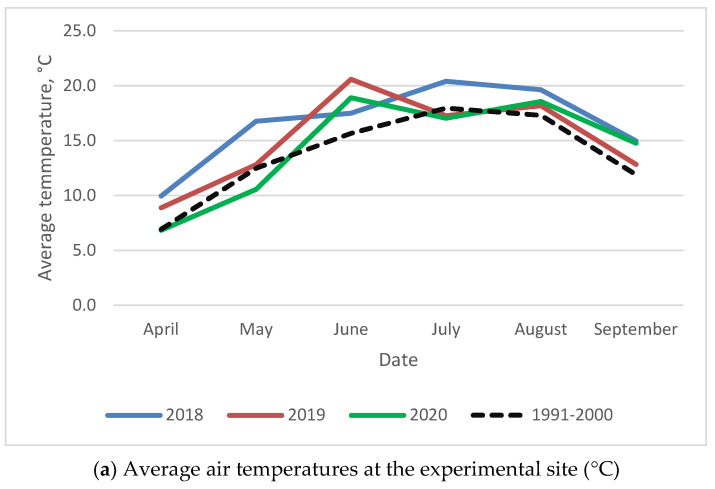
Meteorological conditions from 2018–2020 in Akademija.

**Figure 2 plants-11-02936-f002:**
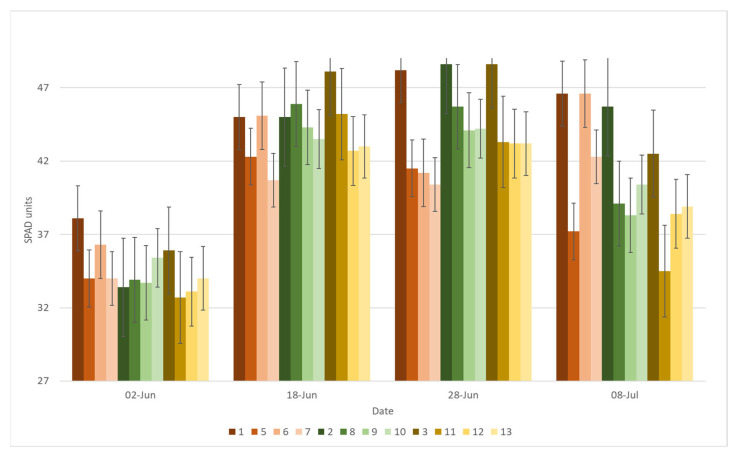
SPAD value of field pea in different treatments, averaged over the three years, depending on the vegetation date. Note: 1, Pea Jūra DS; 2, Pea Egle DS; 3, Pea Lina DS; 4, Oat Viva DS; 5, Pea Jūra DS 50% + 50% oat; 6, Pea Jūra DS 60% pea + 40% oat; 7, Pea Jūra DS 70% + 30% oat; 8, Pea Egle DS 50% + 50% oat; 9, Pea Egle DS 60% pea + 40% oat; 10, Pea Egle DS 70% + 30% oat; 11, Pea Lina DS 50% + 50% oat; 12, Pea Lina DS 60% pea + 40% oat; 13, Pea Lina DS 70% + 30% oat.

**Figure 3 plants-11-02936-f003:**
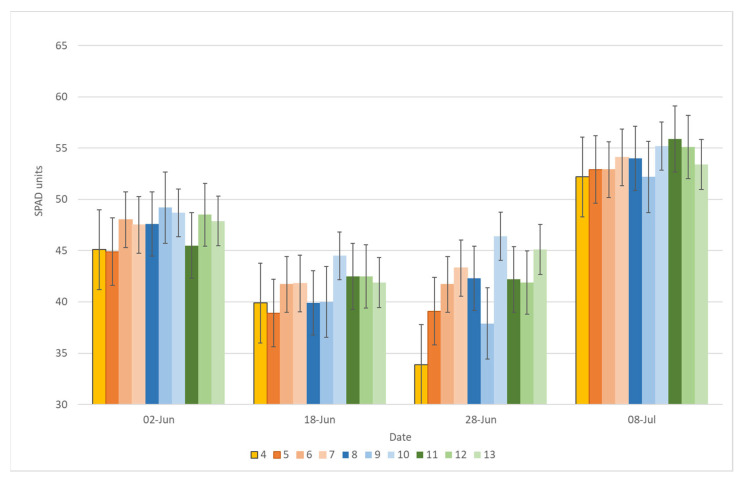
SPAD value of oat in different mixed cropping treatments, averaged over the three years, depending on the vegetation date. Note: 1, Pea Jūra DS; 2, Pea Egle DS; 3, Pea Lina DS; 4, Oat Viva DS; 5, Pea Jūra DS 50% + 50% oat; 6, Pea Jūra DS 60% pea + 40% oat; 7, Pea Jūra DS 70% + 30% oat; 8, Pea Egle DS 50% + 50% oat; 9, Pea Egle DS 60% pea + 40% oat; 10, Pea Egle DS 70% + 30% oat; 11, Pea Lina DS 50% + 50% oat; 12, Pea Lina DS 60% pea + 40% oat; 13, Pea Lina DS 70% + 30% oat.

**Figure 4 plants-11-02936-f004:**
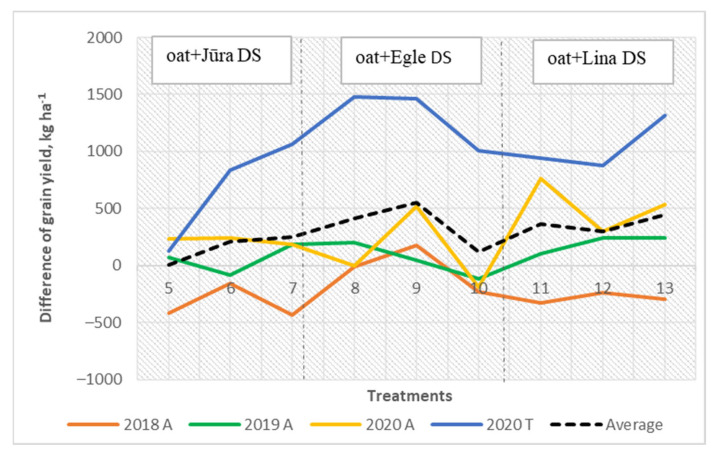
The combined total grain yield: difference between actual and expected grain yield.

**Figure 5 plants-11-02936-f005:**
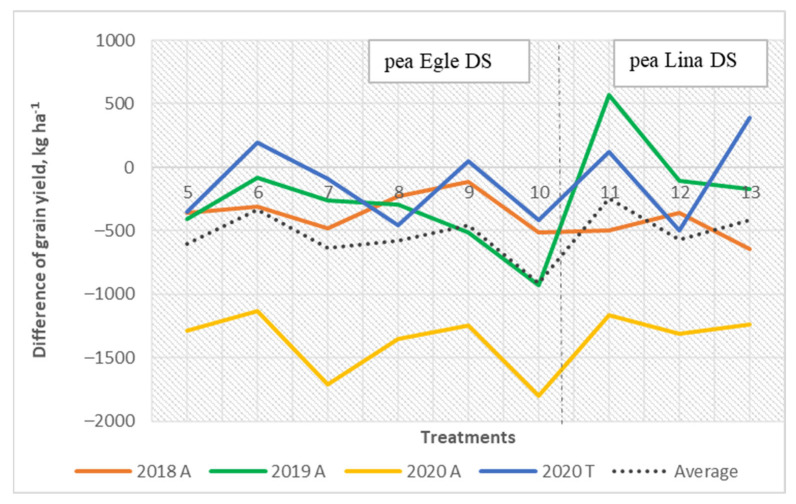
The grain yield of pea in the mixed cropping: difference between actual and expected grain yield.

**Figure 6 plants-11-02936-f006:**
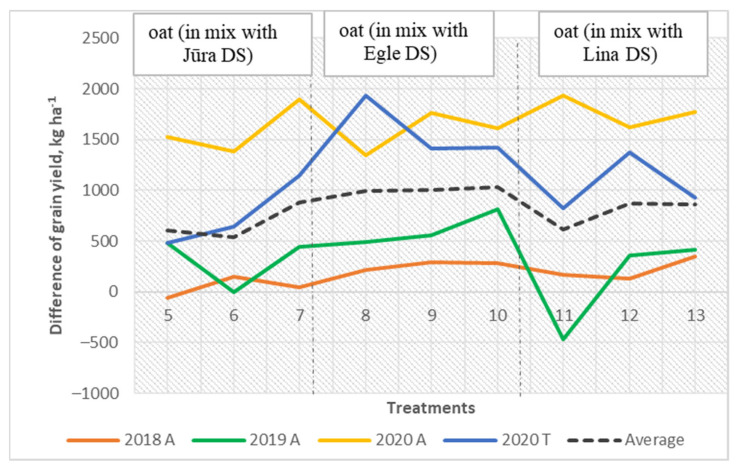
The grain yield of oat in the mixtures: difference between actual and expected grain yield.

**Table 1 plants-11-02936-t001:** The crops’ 1000-kernel weights and protein content in the grains, averaged over the three years, depending on the pea sowing rate and the different cultivars.

Treatment No.	Ratio Pea:Oat	Variety of Pea	1000-Kernel Weight of Pea, g	1000-Kernel Weight of Oat, g	Protein: Pea, %	Protein: Oat, %
The effect of treatment						
1	1:0	Jūra DS	266 ab	-	22.3 cd	-
2	1:0	Egle DS	264 abc	-	23.6 b	-
3	1:0	Lina DS	244 bcde	-	22.6 c	-
4	0:1	-	-	33.6 abc	-	12.1 b
5	1:1	Jūra DS	252 abcd	33.8 abc	22.0 cd	12.8 ab
6	3:2	Jūra DS	252 abcd	34.1 ab	21.6 d	13.0 ab
7	7:3	Jūra DS	252 abcd	34.3 a	22.0 cd	13.1 ab
8	1:1	Egle DS	269 ab	33.4 bc	24.1 ab	13.1 ab
9	3:2	Egle DS	273 a	33.3 c	24.5 a	13.4 a
10	7:3	Egle DS	268 ab	33.3 c	23.9 b	13.2 ab
11	1:1	Lina DS	226 e	33.5 abc	22.2 cd	12.9 ab
12	3:2	Lina DS	233 de	33.7 abc	22.0 cd	12.9 ab
13	7:3	Lina DS	233 de	33.7 abc	22.1 cd	13.0 ab
*Probability*			*<0.001*	*<0.001*	*<0.001*	*<0.001*
The effect of pea and oat ratio						
	1:0	Pea (monocrop)	258 a	*-*	*22.8* a	*-*
	0:1	Oat (monocrop)	-	*33.6* a	*-*	*12.1* b
	1:1	Pea/oat (mixed)	249 a	*33.6* a	*22.8* a	*12.9* a
	3:2	Pea/oat (mixed)	253 a	*33.7* a	*22.7* a	*13.1* a
	7:3	Pea/oat (mixed)	253 a	*33.8* a	*22.6* a	*13.1* a
*Probability*			0.5823	*<0.001*	*0.6768*	*<0.001*
The effect of pea variety in the mixture or monocrop						
		Oat (monocrop)	-	*33.6* b	*-*	*12.1* b
		Jūra DS	256 a	*34.1* a	*22.0* b	*12.9* ab
		Egle DS	269 a	*33.3* ab	*24.0* a	*13.2* a
		Lina DS	235 b	*33.7* ab	*22.2* b	*12.9* ab
*Probability*			*<0.001*		*<0.001*	
*Pea variety × seed ratio*			0.7381	*0.5225*	*0.0099*	*0.9963*

Note. Different letters indicate significant (*p* < 0.05) differences between the means.

**Table 2 plants-11-02936-t002:** Crop height and number of oat grains per ear, averaged across the three years, depending on the pea sowing rate and the different cultivars.

Treatment No.	Ratio Pea:Oat	Variety of Pea	Height of Pea, cm	Height of Oat, cm	Oat Grain Number Per Ear
The effect of treatment					
1	1:0	Jūra DS	73.5 bc	-	-
2	1:0	Egle DS	98.3 a	-	-
3	1:0	Lina DS	79.8 b	-	-
4	0:1	-	-	88.5 a	37.5 b
5	1:1	Jūra DS	57.9 e	96.5 a	60.3 a
6	3:2	Jūra DS	60.2 e	94.6 a	58.3 a
7	7:3	Jūra DS	61.8 e	95.9 a	50.0 ab
8	1:1	Egle DS	70.0 cd	99.5 a	59.5 a
9	3:2	Egle DS	74.1 bc	100.4 a	65.2 a
10	7:3	Egle DS	75.2 bc	98.6 a	62.2 a
11	1:1	Lina DS	58.3 e	92.5 a	62.6 a
12	3:2	Lina DS	61.0 e	95.3 a	60.9 a
13	7:3	Lina DS	64.1 de	86.9 a	55.8 a
*Probability*			*<0.001*		*<0.001*
The effect of pea and oat ratio					
	1:0		83.8 a	*-*	*-*
	0:1		-	*88.5* a	*37.5* b
	1:1		62.0 c	*96.2* a	*60.8* a
	3:2		65.1 bc	*96.8* a	*61.5* a
	7:3		67.0 b	*93.8* a	*56.0* a
*Probability*			*<0.001*		*<0.001*
The effect of pea variety in the mixture or monoculture					
		Oat (monocrop)	-	*88.5* a	*37.5* a
		Jūra DS	63.3 b	*71.8* b	*42.2* a
		Egle DS	79.4 a	*74.6* b	*46.7* a
		Lina DS	65.8 b	*68.7* b	*44.8* a
*Probability*			*<0.001*	*<0.001*	*0.3876*
*Pea variety × seed ratio*			0.1221	*0.8468*	*0.8361*

Note. Different letters indicate significant (*p* < 0.05) differences between the means.

**Table 3 plants-11-02936-t003:** The biomass and straw weight of crops per unit of area (g m^−2^), averaged across the three years, depending on the pea sowing rate and the different cultivars.

Treatment No.	Ratio Pea:Oat	Variety of Pea	Total Biomass, g	Biomass: Pea, g	Biomass: Oat, g
	The effect of treatment				
1	1:0	Jūra DS	555 b	555 c	-
2	1:0	Egle DS	730 ab	730 a	-
3	1:0	Lina DS	648 ab	648 b	-
4	0:1	-	736 ab	-	736 a
5	1:1	Jūra DS	758 a	141 e	617 ab
6	3:2	Jūra DS	758 a	241 d	517 bc
7	7:3	Jūra DS	619 ab	205 de	413 c
8	1:1	Egle DS	669 ab	168 de	501 bc
9	3:2	Egle DS	719 ab	226 de	493 bc
10	7:3	Egle DS	692 ab	250 d	442 c
11	1:1	Lina DS	674 ab	181 de	493 bc
12	3:2	Lina DS	716 ab	182 de	534 bc
13	7:3	Lina DS	675 ab	128 de	457 c
*Probability*			0.3578	*<0.001*	*<0.001*
*Average*			688.4	*288.1*	*400.3*
	The effect of pea and oat ratio				
	1:0		645 a	*645* a	*-*
	0:1		736 a	*-*	*736* a
	1:1		701 a	*163* c	*537* b
	3:2		731 a	*216* b	*514* bc
	7:3		662 a	*224* b	*437* c
*Probability*			0.2228	*<0.001*	*<0.001*
*Average*					
	The effect of pea variety in the mixture or monoculture				
		Oat (monocrop)	736 a	*-*	*736* a
		Jūra DS	672 a	*285* b	*387* b
		Egle DS	703 a	*344* a	*359* b
		Lina DS	678 a	*307* ab	*371* b
*Probability*			0.6846	*<0.001*	*<0.001*
*Pea variety × seed ratio*			0.2952	*0.0155*	*0.4944*

Note. Different letters indicate significant (*p* < 0.05) differences between the means.

**Table 4 plants-11-02936-t004:** The grain yield of crops during three years in four experiments, depending on the pea sowing rate and different cultivars.

Treatment No.	Ratio Pea:Oat	Variety of Pea	2018 A*	2019 A*	2020 A*	2020 T*	Average
1	1:0	Jūra DS	1639 abcd	2403 e	4169 e	1938 e	2537 b
2	1:0	Egle DS	1707 abcd	2958 d	4863 cd	2892 cd	3105 ab
3	1:0	Lina DS	1915 abc	2207 e	4037 e	2002 e	2540 b
4	0:1	-	2177 a	3905 a	5396 abc	3489 bcd	3742 a
5	1:1	Jūra DS	1491 cd	3225 cd	5017 abcd	2841 d	3144 ab
6	3:2	Jūra DS	1698 abcd	2921 d	4906 bcd	3395 bcd	3230 ab
7	7:3	Jūra DS	1367 d	3037 cd	4723 d	3464 bcd	3148 ab
8	1:1	Egle DS	1936 abc	3633 ab	5126 abcd	4669 a	3841 a
9	3:2	Egle DS	2072 ab	3383 bc	5594 a	4594 a	3911 a
10	7:3	Egle DS	1617 bcd	3130 cd	4838 cd	4076 ab	3415 ab
11	1:1	Lina DS	1716 abcd	3159 cd	5479 ab	3687 bc	3510 ab
12	3:2	Lina DS	1787 abcd	3132 cd	4884 bcd	3477 bcd	3320 ab
13	7:3	Lina DS	1701 abcd	2959 d	4928 bcd	4038 ab	3351 ab
*Probability*		*Probability*	0.068	<*0.001*	<*0.001*	<0.001	0.0434
*Average*		*Average*	1756	*3081*	*4920*	*3439*	*3292*

Note: A*, Akademija experimental site; T*, Tiskūnai experimental site. Different letters indicate significant (*p* < 0.05) differences between the means.

**Table 5 plants-11-02936-t005:** Land equivalent ratio for grain (LER), relative crowding coefficient of Crop a intercropped with Crop b (k), aggressivity (A) and competitive ratio (CR) of oat–pea mixed crops as affected by the ratio (%), averaged over the three years.

Treatment No.	Ratio Pea:Oat	Variety of Pea	LER	k: Oat	k: Pea	A: Oat	A: Pea	CR: Oat	CR: Pea
		The effect of treatment							
1	1:0	Jūra DS	1.00 ab	-	-	-	-	-	-
2	1:0	Egle DS	1.00 ab	-	-	-	-	-	-
3	1:0	Lina DS	1.00 ab	-	-	-	-	-	-
4	0:1	-	1.00 ab	-	-	-	-	-	-
5	1:1	Jūra DS	0.91 b	2.29 a	0.43 ab	0.70 bcd	−0.70 abc	2.99 a	0.51 bc
6	3:2	Jūra DS	1.03 ab	2.25 a	0.19 ab	0.48 cd	−0.48 ab	2.09 a	0.72 ab
7	7:3	Jūra DS	0.99 ab	2.87 a	0.56 ab	0.99 ab	−0.99 cd	2.85 a	0.46 bc
8	1:1	Egle DS	1.09 a	2.51 a	0.55 ab	0.85 abcd	−0.85 abcd	4.10 a	0.48 bc
9	3:2	Egle DS	1.13 a	1.86 a	0.89 a	0.83 abcd	−0.83 abcd	2.30 a	0.52 bc
10	7:3	Egle DS	0.98 ab	5.70 a	0.41 ab	1.27 a	−1.27 d	4.03 a	0.35 c
11	1:1	Lina DS	1.07 a	3.25 a	1.20 a	0.41 d	−0.41 a	2.18 a	0.87 a
12	3:2	Lina DS	1.00 ab	3.30 a	0.50 ab	0.88 abcd	−0.88 abcd	2.78 a	0.48 bc
13	7:3	Lina DS	1.08 a	2.88 a	−0.50 b	0.91 abc	−0.91 bcd	2.60 a	0.52 bc
*Probability*			0.1408	*0.5887*	*0.1354*	*0.0052*	*0.0052*	*0.5037*	*0.0093*
*Average*			1.02	*2.99*	*0.469*	*0.813*	*−0.813*	*2.881*	*0.549*
		The effect of pea and oat ratio							
	1:0		1.00 a	*-*	*-*	*-*	*-*	*-*	*-*
	0:1		1.00 a	*-*	*-*	*-*	*-*	*-*	*-*
	1:1		1.02 a	2.69 a	0.73 a	0.65 b	−0.65 a	3.09 a	0.62 a
	3:2		1.05 a	3.82 a	0.53 a	0.73 b	−0.73 a	2.39 a	0.57 ab
	7:3		1.02 a	2.47 a	0.16 a	1.06 a	−1.06 b	3.16 a	0.44 b
*Probability*			0.5756	*0.3628*	*0.1686*	*0.0034*	*0.0034*	*0.4014*	*0.0648*
		The effect of pea variety in the mixture or monocrop							
		Oat (monocrop)	1.00 a	*-*	*-*	*-*	*-*	*-*	*-*
		Jūra DS	0.98 a	2.47 a	0.39 a	0.72 b	−0.72 a	2.65 a	0.56 ab
		Egle DS	1.05 a	3.36 a	0.62 a	0.98 a	−0.98 b	3.48 a	0.45 b
		Lina DS	1.04 a	3.15 a	0.40 a	0.72 b	−0.73 a	2.52 a	0.62 a
*Probability*			*0.1031*	*0.6601*	*0.7094*	*0.0646*	*0.0646*	*0.2588*	*0.0813*
*Pea variety x seed ratio*			0.1408	*0.4571*	*0.0852*	*0.2268*	*0.2268*	*0.5974*	*0.0345*

Note. Different letters indicate significant (*p* < 0.05) differences between the means.

## Data Availability

All data included in the main text.

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
