# Peer review of "Effects of Pea (*Pisum sativum* L.) Cultivars for Mixed Cropping with Oats (*Avena sativa* L.) on Yield and Competition Indices in an Organic Production System"

_plants, 2022, doi:10.3390/plants11212936_

Round 1
Reviewer 1 Report
The article is devoted to the study of the influence of pea and oat plants on each other when grown together.
The article is completely agronomic and should be submitted to an agronomic journal.
The setting of the experiment is very complex and intricate.
One-factor experiment is excluded.
Novelty is doubtful, these questions have been studied for many years since the last century.
Statistical processing cannot be used one-way Anova because it's a lot of factors.
Author Response
Dear Colleague,
Thank you for your time and comments.
Due to Novelty of the research, we added in the introduction overview. The investigation of pea varieties in a mixed cropping system can show that variety diversification may increase yield and promote microbial interactions by affecting the soil, plant health and broader ecosystem functions. The implementation of the Green Deal program, reducing the use of fertilizers, pesticides, energy costs will stimulate (in part) the cultivation of mixed crops. Therefore, improving the practice of growing mixed crops will make these crops more attractive and at the same time gain greater confidence from growers.
The methods part was improved.
According to statistic comment: There are two factors in our experiment. As you may see in the manuscripts, that we test the interaction of both factors, and in most cases the interaction is insignificant, which led us to run one-way Anova. In this way the treatment is as a combination of two factors or just simply preferred management option. Only some data had significant interactions for factors, for which in the tables we give also the numbers of two-factorial analyses. To arrange those in different style tables would take to much space, so here we clearly show where the interactions occur and assume that scientific readers will understand the tables and statistics.
Kind regards,
Authors

Reviewer 2 Report
The authors studied the Effects of pea (Pisum sativum L.) cultivars for mixed cropping with oats (Avena sativa L.) on yield and competition indices in an organic production system. The work is well structured, presented and the obtained results has a practical value to be used in organic mixed cropping systems.
However, there are some corrections that are needed:
1. In Material and method section:
a) along with average rainfall and temperature I recommend indicate annual rainfall and temperature for each year separately as well. As we know, sometimes climatic characteristics are very differed from year to year (from drought in one year to flooding the next year, etc.) and it has a big influence on plant growth and quality.
b) Include Description of which methods were used to assess photosynthetic activity (SPAD value), protein content.
2. Did you assess the biological N fixation? If no, from conclusion part “ … which was mainly influenced by the effects of biological nitrogen fixed by pea….” should be removed and may be mentioned on Discussion part.
3. Author contribution section must be included at the end of the article.
4. References should be numbered in order of appearance and indicated by a numeral or numerals in square brackets—e.g., [1] or [2,3], or [4–6]. See the end of the Template document for further details on references. https://www.mdpi.com/journal/plants/instructions
5. I would like to recommend to include some information about changes in protein content in Abstract as well.
In the attachment, the article with my comments. Yellow color indicates places requiring clarifications.
Best wishes

Author Response
Dear Colleague,
Thank you for your time and valuable comments.
- In Material and method section:
- a) along with average rainfall and temperature I recommend indicate annual rainfall and temperature for each year separately as well. As we know, sometimes climatic characteristics are very differed from year to year (from drought in one year to flooding the next year, etc.) and it has a big influence on plant growth and quality.
Added weather conditions.
- b) Include Description of which methods were used to assess photosynthetic activity (SPAD value), protein content.
Description of SPAD value added.
- Did you assess the biological N fixation? If no, from conclusion part “ … which was mainly influenced by the effects of biological nitrogen fixed by pea….” should be removed and may be mentioned on Discussion part.
The sentence removed from Conclusion part and mentioned into Discussion part.
- Author contribution section must be included at the end of the article.
Authors contribution section is added in the end of article.
- References should be numbered in order of appearance and indicated by a numeral or numerals in square brackets—e.g., [1] or [2,3], or [4–6]. See the end of the Template document for further details on references. https://www.mdpi.com/journal/plants/instructions
References is numbered according to the journal requirements.
- I would like to recommend to include some information about changes in protein content in Abstract as well.
Information about the protein content was added in Abstract.
In the attachment, the article with my comments. Yellow color indicates places requiring clarifications.
We try to add all clarifications.
Kind regards,
Authors

Reviewer 3 Report
This article studied Effects of Pea (Pisum sativum L.) Cultivars for Mixed Cropping with Oats (Avena sativa L.) on Yield and Competition Indices in an Organic Production System. The benefits of legumes are well known therefore mix cropping may help in the production of the legumes. Before recommending this article for publication, there are some shortcomings for that should be resolve.
General comments
I would suggest revising the MS with an expert English writer it will help to convey clear information to readers of this journal.
Abstract
Write abbreviation “Egle DS”
The authors must show the methods of treatment application on plants.
Quantitative results are missing in the abstract.
How the authors quantified or got the readings?
I strongly recommend writing small and clear sentences.
Introduction
Add commercial and industrial importance of both the crops.
Significance of mixed crops systems.
How they respond to weeds etc.
“In order to make the most efficient use of the agrobiological resources in agriculture and food, it is necessary to find ways to reduce synthetic pesticides and make more efficient use of the potential value of grain legumes” can be change in to potential value of crops by using organic resources. The following studies could be cited. DOI: 10.1016/j.micpath.2020.103966,
https://doi.org/10.1007/s10534-022-00417-1,
The result and discussion are well presented but grammatical mistakes must be revised by the authors.
Conclusion is well justified. The authors should discuss some points for the future studies molecular level studies are required to know about the involved mechanism and improvement of mixed crop system.
Author Response
Dear Colleague,
Thank you for your time and valuable comments.
General comments
I would suggest revising the MS with an expert English writer it will help to convey clear information to readers of this journal.
Abstract
Write abbreviation “Egle DS”
The authors must show the methods of treatment application on plants.
Quantitative results are missing in the abstract.
How the authors quantified or got the readings?
I strongly recommend writing small and clear sentences.
All recommendations were done in the Abstract part. Expert of English writer checked the manuscript.
Introduction
Add commercial and industrial importance of both the crops.
Significance of mixed crops systems.
How they respond to weeds etc.
In the Introduction part was don corrections according to recommendations.
“In order to make the most efficient use of the agrobiological resources in agriculture and food, it is necessary to find ways to reduce synthetic pesticides and make more efficient use of the potential value of grain legumes” can be change in to potential value of crops by using organic resources. The following studies could be cited. DOI: 10.1016/j.micpath.2020.103966,
https://doi.org/10.1007/s10534-022-00417-1,
The result and discussion are well presented but grammatical mistakes must be revised by the authors.
Expert of English writer checked.
Conclusion is well justified. The authors should discuss some points for the future studies molecular level studies are required to know about the involved mechanism and improvement of mixed crop system.
We consider suggestion in the future studies.
Kind regards,
Authors

Reviewer 4 Report
Reviewer’s comments on manuscript titled “Effects of Pea (Pisum sativum L.) Cultivars for Mixed Cropping with Oats (Avena sativa L.) on Yield and Competition Indices in an Organic Production System”.
GENERAL COMMENTS
The authors have presented a research quantifying yield and competition indices as impacted by multi-cropping of pea with oats in different ratios. While such studies are not novel and there are a lot of studies on similar research topics even involving the same two crops, it may be still relevant for the scientific community as it involves new pea varieties and is acceptable for publication in the journal. However, there are several concerns which need to be addressed before making this paper acceptable for publication in the journal. Following are some of the general comments, for more detailed comments, please see the attached file.
General comment 1: The introduction section must be improved by adding relevant studies describing similar studies and knowledge gap which the authors would like to address. Additionally, the introduction section lacks a strong problem statement.
General comment 2: The manuscript needs to be moderately edited for language/grammar issues. I have done some edits but may have missed many places for sure.
General comment 3: The material and methods section lacks some key details. For example, the authors have presented a section on SPAD in results, however, there is no such parameter described in M&M section. There are several other key information missing. Please see the attached file for the details.
General comment 4: The results section needs to be heavily edited. Most of the tables are confusing and needs to be broken down into several tables. One of the most concerning parts is the interpretation of results. The authors have indicated differences at several instances, however, there are no significant differences. One set of results on SPAD is presented which is incomprehensible as there is no such parameter explained in the M&M section. For other specific comments, please see the attached file.
General comment 5: The discussion section needs to be edited with discussion around the key results. This section contains some general information which may be a part of the introduction section and not the discussion. See the attached file for detailed comments.

Author Response
Dear Colleague,
Thank you for your time and valuable comments.
GENERAL COMMENTS
The authors have presented a research quantifying yield and competition indices as impacted by multi-cropping of pea with oats in different ratios. While such studies are not novel and there are a lot of studies on similar research topics even involving the same two crops, it may be still relevant for the scientific community as it involves new pea varieties and is acceptable for publication in the journal. However, there are several concerns which need to be addressed before making this paper acceptable for publication in the journal. Following are some of the general comments, for more detailed comments, please see the attached file.
The detailed comments were followed by authors and corrected in the manuscript.
General comment 1: The introduction section must be improved by adding relevant studies describing similar studies and knowledge gap which the authors would like to address. Additionally, the introduction section lacks a strong problem statement.
In the Introduction part was done corrections according to recommendations.
General comment 2: The manuscript needs to be moderately edited for language/grammar issues. I have done some edits but may have missed many places for sure.
Thank you for edits. The manuscript was checked by Expert of English writer.
General comment 3: The material and methods section lacks some key details. For example, the authors have presented a section on SPAD in results, however, there is no such parameter described in M&M section. There are several other key information missing. Please see the attached file for the details.
Description of SPAD value added and described in M&M section.
General comment 4: The results section needs to be heavily edited. Most of the tables are confusing and needs to be broken down into several tables. One of the most concerning parts is the interpretation of results. The authors have indicated differences at several instances, however, there are no significant differences. One set of results on SPAD is presented which is incomprehensible as there is no such parameter explained in the M&M section. For other specific comments, please see the attached file.
According to the specific comments in the attached file, the results section and M&M was improved.
General comment 5: The discussion section needs to be edited with discussion around the key results. This section contains some general information which may be a part of the introduction section and not the discussion. See the attached file for detailed comments.
According to the specific comments in the attached file, text was improved.
Kind regars,
Authors

Round 2
Reviewer 1 Report
The article is devoted to the study of the influence of pea and oat plants on each other when grown together.
Novelty is doubtful. these questions have been studied for many years since the last century.
It is necessary to indicate whether additional feeding or any nutrient medium was used.
Author Response
Dear Collegue,
Thank you for your affort and comments to seek that manuscript would be be improved.
Comments and Suggestions:
- Novelty is doubtful these questions have been studied for many years since the last century.
Response: The investigation focused on new pea varieties in a mixed cropping system. It is imprtant to explore that variety diversification may increase yield and promote microbial interactions by affecting the soil, plant health and broader ecosystem functions, especialy in organic farming. The selection of pea genotypes with comparable phenology but contrasting stature and growth when intercropped with cereals in breeding programmes helped to determine pea proportion.
2. It is necessary to indicate whether additional feeding or any nutrient medium was used.
Response: The investigation was carried out in an organic farming system therefore any nutrient application was not used. This indicated in the methods part, naw.
Kind wishes,
Authors
Reviewer 4 Report
Dear Authors,
Thanks for submitting the revised manuscript which is in a much better shape. I have a few suggestions which are highlighted in the attached version of the manuscript.
One thing I would like to highlight is that the authors should provide a justification if they don't chose to make changes based on a reviewer comment. I had provided a number of comments in the previous version which were neither incorporated nor any justification was provided.
I have highlighted some of those key comments in the attached version.
Thanks

Author Response
Dear Collegue,
Thank you for your affort and comments to seek that manuscript would be be improved. We follow all remarks in the manuscript.
Some justification:
According to literature numers:
Citation is corrected according to guidelines.
According to wheather data:
There were 0.00 mm precipitation at the experimental site in April 2019. So empty column for it.
according to Table 1:
It is proposed to split Table 1, but the tables are the same structure throughout the manuscript. Therefore, we believe that changing the structure of first table, it would lead to unevenness and make it difficult for the reader to follow all results due to other tables would be in different shape.
Kind regards,
Authors